# A Food Waste-Derived Organic Liquid Fertiliser for Sustainable Hydroponic Cultivation of Lettuce, Cucumber and Cherry Tomato

**DOI:** 10.3390/foods12040719

**Published:** 2023-02-07

**Authors:** Zuhaib Siddiqui, Dharmappa Hagare, Min-Hang Liu, Orousa Panatta, Tanveer Hussain, Sheeraz Memon, Amber Noorani, Zhong-Hua Chen

**Affiliations:** 1School of Engineering, Design and Built Environment, Western Sydney University, Penrith, NSW 2751, Australia; 2School of Science and Technology, Faculty of Science, Agriculture, Business and Law, University of New England, Armidale, NSW 2350, Australia; 3Institute of Environmental Engineering, Mehran University of Engineering and Technology, Jamshoro 76062, Pakistan; 4Department of Biochemistry, Faculty of Basic Medical Science, Jinnah Sindh Medical University, Karachi 75510, Pakistan; 5School of Science, Western Sydney University, Richmond, NSW 2753, Australia

**Keywords:** sustainable agriculture, macronutrients, cations, FoodLift, circular economy, sustainable waste management

## Abstract

We previously reported a sustainable food waste management approach to produce an acceptable organic liquid fertiliser for recycling food waste called “FoodLift.” This study follows our previous work to evaluate the macronutrients and cation concentrations in harvested structural parts of lettuce, cucumber, and cherry tomatoes produced using food waste-derived liquid fertiliser (FoodLift) and compare them against commercial liquid fertiliser (CLF) under hydroponic conditions. N and P concentrations in the structural parts of lettuce and the fruit and plant structural parts of cucumber appear to be similar between FoodLift and CLF (*p* > 0.05), with significantly different N concentrations in the various parts of cherry tomato plants (*p* < 0.05). For lettuce, N and P content varied from 50 to 260 g/kg and 11 to 88 g/kg, respectively. For cucumber and cherry tomato plants, N and P concentrations ranged from 1 to 36 g/kg and 4 to 33 g/kg, respectively. FoodLift was not effective as a nutrient source for growing cherry tomatoes. Moreover, the cation (K, Ca, and Mg) concentrations appear to significantly differ between FoodLift and CLF grown plants (*p* < 0.05). For example, for cucumber, Ca content varied from 2 to 18 g/kg for FoodLift grown plants while Ca in CLF-grown cucumber plants ranged from 2 to 28 g/kg. Overall, as suggested in our previous work, FoodLift has the potential to replace CLF in hydroponic systems for lettuce and cucumber. This will lead to sustainable food production, recycling of food waste to produce liquid fertiliser, and will promote a circular economy in nutrient management.

## 1. Introduction

Food waste (FW) refers to the biodegradable organic waste derived from different sectors, such as unmarketable farm produce, by-products from the food processing industry, hospitality, and households. Pre-cooked and leftover foods also possibly produce biodegradable waste. The issue of food waste has become a global concern and is recommended to emerge as a priority in political agendas globally [1,2]. The Food and Agriculture Organisation (FAO) has reported that approximately one-third of food produced in the world ends up as food waste, which equals approximately 1300 × 10^6^ tonne/yr [3]. This amount of wasted food can sufficiently feed 1.5 billion hungry people in some poor communities. Additionally, it has been reported that approximately 12.5% of the global population (one billion) are malnourished due to deficiencies in daily food intake [4]. Thus, recycling of food waste can effectively circumvent global malnutrition and hunger. Recycling of food waste becomes even more significant in the light of the increasing world population and acceleration of climate change.

Food waste also causes a significant carbon footprint, which contributes to greenhouse gas emissions by expelling approximately 3300 million tonnes of CO_2_ into the atmosphere per year. Typically, food waste can be dumped in an open area, incinerated, or other optional treatments that may cause some environmental and severe health issues [5,6]. Accordingly, waste-to-energy systems, including landfill and anaerobic digestion, have been recommended as alternative disposal processes for food waste. These methods contain minimal risks to the soil, air, and water [7,8]. The unique advantage of the anaerobic digestion waste system is that it generates electricity from the biogas, and natural gas is used as a resource to supply heat and power units [9]. Biogas is a cleaner energy than other fossil fuels, has a low discharge level of greenhouse emissions, and can be generated from food waste [10].

Hydroponics in greenhouses and indoor cropping facilities is an efficient method utilised to grow a range of fruits and vegetables with high yield and quality [11,12,13]. This technique relies on water and mineral nutrients to grow plants under soilless conditions [14]. The hydroponic method as an alternative plant production technique poses various advantages, such as faster plant growth, shorter crop cycles, easy control of the composition of nutrients, contamination-free soil, high plant quality, and fresh produce throughout the year [15]. Furthermore, high-tech protected cropping is characterised by high quality and yield of the crop irrespective of climate, weather, and soil conditions [16,17], increased reliability of fresh produce supply, and increased choices/options for alternative packaging and presentation with enhanced shelf life [18]. As a result, significant reductions in traditional agricultural land areas have been replaced with protected cropping in many countries [19]. This has evolved from very simple polytunnels to complex high-tech greenhouses in European countries and expanded across the world [20,21,22].

Lettuce (*Lactuca sativa* L.) is a green leafy plant from the Asteraceae family and is generally consumed in salad mixes. Lettuce is a vegetable that contains a high level of nutrition and is a good source of fibre, minerals, and vitamin C [23]. The cool season is the most suitable condition for growing lettuce, as temperatures range from 7–24 °C [24]. Currently, lettuce consumption is surging due to increasing demand for healthy leafy vegetables from the large global population. Cucumber (*Cucumis sativus* L.) is one of the oldest horticultural crops, domesticated over five thousand years ago, potentially originating in India [25]. Cucumber is thermophilic and can grow in countries located in temperate zones, and a temperature higher than 20 °C can give the maximum yield [26]. It was reported that hydroponic cucumber in Australia could grow by relying on recycled water and nutrients [27], reducing the cost of hydroponic production [28]. Tomato (*Solanum lycopersicum* L.) is one of the most economically important vegetables because it contains a high nutritional value [29]. However, it is still expensive, especially cherry tomatoes [30]. The economic return of tomato production is relatively fast with a high commercial value, especially for hydroponic tomatoes in winter. Nevertheless, tomatoes require a high level of nutrition. The appropriate amount of minerals in the hydroponic nutrient solution can enhance the yield of tomatoes, and the nutrient concentration must be adjusted constantly for different stages of growth [31].

A major limitation in crop cultivation is inadequate land area for planting enough food to feed the booming global population [32]. It has been stated that the effluent, which is discharged from the anaerobic digestion process of poultry manure, can be utilised as liquid fertiliser to grow hydroponic lettuce with a higher growth rate than using commercial fertiliser [24]. Chew et al. [33] reported the advantages of using waste biomass as an organic fertiliser. Recently, new alternative food waste utilisation has been developed to convert food waste into organic liquid fertiliser (FoodLift) and animal feed [33,34]. Siddiqui et al. [35] reported the yields of lettuce and cucumber using FoodLift as liquid fertiliser. However, this work did not include the nutrient concentrations in the fruits and the different structural parts of the plants. Table 1 summarises the nutrient concentrations in the various parts of the plant structures. The aim of this study is to compare the nutrient concentrations in the different parts of plant structures grown using FoodLift with the corresponding plant structures grown using CLF. This will help to identify any major differences in the nutrient concentrations between the plants/fruits grown using FoodLift and CLF.

In this study, the harvested products of lettuce, cucumber, and tomato, such as fruit, roots, stem, stalk, and leaves were analysed for both macronutrients and cations. This was undertaken for all plant products produced using both FoodLift and CLF.

## 2. Materials and Methods

### 2.1. Raw Materials

#### 2.1.1. Seedlings

The seedlings of cos lettuce (*Lactuca sativa* L., cultivar fLORIANA), continental cucumber (*Cucumis sativus* L., cultivar fLORIANA), and cherry tomato (*Solanum lycopersicum* L.) were acquired from a local nursery (Bunnings, Penrith, NSW, Australia).

#### 2.1.2. Liquid Fertilisers

An organic liquid fertiliser extracted from food waste, called “FoodLift”, was used in this study. FoodLift was extracted following the steps outlined in Siddiqui et al. [34]. The food waste sources were fruits and vegetables from retail shops. The waste was collected over 3 seasons, namely, winter, spring, and summer [34]. pH, EC, total N, total P, free reactive phosphorus (FRP), dissolved organic carbon (DOC), Ca, Mg, Na, and K parameters are presented in Siddiqui et al. [35].

To compare the yield and nutrient concentrations of produce obtained using FoodLift, experiments were repeated using commercial liquid fertilisers (CLF). CLF used in the experiments was sourced from a commercial hydroponic solution provider [41]. There were two types of CLFs, namely, Hydro Grow and Hydro Bloom. Hydro Grow was used for lettuce for the whole growth cycle. In the case of cucumber and tomato, Hydro Grow was used during the vegetative growth stage and Hydro Bloom was used during the reproductive growth phase to meet the nutritional requirements for flowering and fruiting of the 2 crops. Both FoodLift and CLF nutrient tanks were setup side by side as shown in Figure 1. Table 2 shows the characteristics of nutrient solutions for hydroponics prepared following the Dutch Fest HydroGrow [41] instructions. A more detailed characterisation of FoodLift is provided in Siddiqui et al. [35]. Siddiqui et al. [34,35] reported that the macronutrient (nitrogen and phosphorus) concentrations in FoodLift makes it ideally suited to be used as a hydroponic feed solution.

### 2.2. Experimental Setup

The experimental setup of the greenhouse and bato buckets is shown in Figure 1 and is discussed in Siddiqui et al. [35] in detail. The environmental conditions, such as temperature, humidity, and light intensity for lettuce are detailed in Table 2. The hydroponic experiments for growing lettuce, cucumber, and cherry tomato were carried out for 6, 17, and 10 weeks, respectively. Same experimental setup was used for all 3 trials and after completing each trial, the setup was thoroughly washed with clean water and used to conduct the next set of hydroponic experiments. In addition, Siddiqui et al. [35] reported that the fresh matter (FM) yields of the lettuce plant were similar (FoodLift: 156 g/plant; and CLF: 161 g/plant) for both fertilisers. However, in the case of cucumber, the yield obtained using FoodLift (187 g/plant) was significantly lower than the one obtained using CLF (243 g/plant). Reasons behind this difference were explained in Siddiqui et al. [35]. In the case of cherry tomatoes, the number of tomatoes harvested are presented in Table 3. As can be seen in the table, the cherry tomato yield from FoodLift was significantly low. This was mainly attributed to the low P concentration in the FoodLift nutrient solution. The harvested leaves of lettuce and fruits of cucumber and tomato were analysed for various nutrient concentrations.

### 2.3. Analytical Methods

The chemical analyses of the liquid fertilisers were carried out following the procedure outlined in Siddiqui et al. [35]. Greenhouse temperature and humidity were monitored using a Kestrel 5000 Environmental Meter. Luminescence in the greenhouse was measured using a field scout light sensor reader, Spectrum Technologies, Aurora, IL, USA. Statistical analyses, such as the *t*-test and significance analysis, were performed using Microsoft Excel.

## 3. Results and Discussions

### 3.1. Nutrients

Figure 2 and Appendix A show the average concentration of macronutrients (N and P) in harvested parts of plants, such as fruits, roots, stems, stalks and leaves of lettuce, cucumber, and cherry tomato. Leaves of lettuce and fruits of cucumber and cherry tomato are the edible part of these vegetables. N and P concentrations in the leaves of lettuce and fruits of cucumber, and other plant structural parts grown using FoodLift, were observed to be similar to those of CLF. This is also reflected in the significance analysis (*t*-distribution test) as shown in Table 4. The *p* values were estimated to be more than 0.05 which indicate that the N and P concentrations in the structural parts of both lettuce and cucumber were statistically similar. However, in the case of tomatoes, and in particular the nitrogen concentrations in the structural parts of plants, they appear to differ with respect roots. The N concentration in the roots of cherry tomato plants grown using FoodLift was observed to be almost double the concentration found in the cherry tomato roots grown using CLF (Figure 2 and Appendix A). In addition, there appears to be a slightly significant difference in N concentrations in the stem and stalk of tomato plants grown using FoodLift and CLF. On the other hand, the P concentrations in all of the structural parts of cherry tomatoes grown using both FoodLift and CLF were similar, except in the case of leaves (Figure 2c). The differences in N concentrations for roots, stem, and stalk, and P concentrations for leaves of tomato plants are shown to be significant in Table 4. Furthermore, as explained earlier, the hydroponic trials with cherry tomato plants showed a significant difference between FoodLift and CLF. As shown in Table 3, the yield obtained with the use of FoodLift was significantly lower than the one obtained using CLF. This is because FoodLift may not have all the nutrients required for optimal growth of cherry tomato plants, which have relatively high requirements for different nutrients [28]. The authors propose that the P concentration in FoodLift was significantly lower than the one in CLF. Hence, it may be required to supplement P in FoodLift, if it is required to be used for growing cherry tomatoes. Thus, overall, it can be said that N and P concentrations are similar between the structural parts of FoodLift- and CLF-grown lettuce and cucumber plants. Hence, it is possible to use a FoodLift nutrient solution in place CLF for hydroponic growing of lettuce and cucumber.

There are few studies on the nutrient and cation concentrations of the various structural parts of lettuce. Two studies [36,37] reported the P content in the leaves of lettuce in the range of 5.7–8.6 g/kg. The N concentration in the lettuce leaf was reported as 3 g/kg [36]. Compared with the observed concentrations in this study (250 and 87 g/kg for N and P, respectively), the literature concentrations are very low. This may be due to the different growing conditions which were used in the literature.

Similarly, comparing the N and P concentrations for the cucumber plant obtained in this study (Figure 2) with the literature (Table 1), it can be said that the concentrations are similar. On the other hand, the N and P concentrations for the tomato plant obtained in this study (Figure 2) appear to be significantly low compared with the literature values (Table 1).

The pH of the nutrient solution affects the solubility and thus availability of certain elements, such as iron and phosphorus [14,42]. A moderately low pH (5.5) keeps most ions available in a solution, while a higher pH (>6.5) can cause nutrient deprivation due to nutrient precipitation and depletion [42]. Additionally, there may be an electrochemical burden in moving excess ions across membranes under a high apoplastic pH [43]. The liquid fertiliser solution used in this study was initially designed to maintain pH 5.5 ± 0.3 for lettuce and cucumber, and 6.5 ± 0.3 for cherry tomato (Table 2), following the recommendations of Dutch Fest HydroGrow [41]. This range meets the ideal target pH of 5.8 recommended by Bugbee [42] for hydroponic solutions. The pH of the solution also affects how much energy is expended by the plant to import ions across the cell membrane and the tonoplast against electrochemical gradients. In many cases, protons (H^+^ ions) are used in the active co-transport of ions across membranes. The concentration of H^+^ ions in the nutrient solution affects nutrient uptake and transport [44]. Furthermore, the addition of alkaline-pH of makeup-water disturbs the evapotranspiration, and the added carbonate (CO_3_^2−^) and bicarbonate (HCO_3_^−^) may exceed the system’s usage. When this is the case, the pH will rise and reduce the availability of elements such as iron, potassium, and phosphorus. Therefore, the pH was regularly adjusted in the current hydroponic experiments in the range shown in Table 2.

### 3.2. Cations

In addition to macronutrients (N and P), major cations, such as calcium (Ca), magnesium (Mg), and potassium (K) in the structural parts of harvested plants were analysed. Figure 3 and Appendix A show the average concentration of cations Ca, Mg, and K in harvested parts of plants, namely the fruit, roots, stem, stalk, and leaves of lettuce (Figure 3a), cucumber (Figure 3b), and cherry tomato (Figure 3c). In the case of lettuce, calcium concentrations in different parts of the plant grown using FoodLift appear to be significantly different to the calcium concentrations observed in the corresponding part of the plant grown using CLF (Figure 3a and Table 4). On the other hand, in the case of cucumber and cherry tomato, the calcium concentrations differ between the plants grown using FoodLift and CLF, only for certain structural parts of the plants (Table 4). However, there was no easily discernible trend in the calcium concentrations. The main differences were in roots. In addition to the roots, in the case of cucumber, there was a highly significant difference in calcium concentration in the leaves of the cucumber plants grown using FoodLift and CLF.

As far as potassium (Figure 3 and Appendix A) is concerned, its concentrations in different structural parts of lettuce, cucumber, and tomato plants differ significantly between the plants grown using FoodLift and CLF (Figure 3). This is also reflected in the *p*-values as given in Table 4. Some differences, such as the ones between leaves, stems, and stalks in the case of lettuce, roots in the case of cucumber, and fruits, stems, and stalks in the case of tomatoes were found to be highly significant. However, again, there are no definite trends. Results do not indicate preference to any particular fertiliser in terms of uptake of cations by the plants. With respect to magnesium (Figure 3 and Appendix A), there appears to be extremely significant differences in the Mg concentrations in the plants grown using Foodlift and CLF (Table 4).

The above results for cation concentrations In various structural parts of the lettuce, cucumber, and tomatoes indicate that the fertiliser type can influence the uptake of cations. This may be due to the different concentrations of the cations in the nutrient solution (Table 2) and various physiological parameters of the plant. This needs to be further investigated.

There were no literature data to compare Ca, Mg, and K concentrations in lettuce. However, there were some literature data on Ca, Mg, and K concentrations for cucumber and cherry tomato plants (Table 1). Comparing Ca concentrations obtained in this study (Figure 3) with that of the literature (Table 1), it can be concluded that the Ca concentrations (1–35 g/kg) were generally lowers than those reported in the literature (2–50 g/kg). In the case of Mg concentrations obtained in this study (1–18 g/kg) were similar to that of literature (2–15 g/kg). On the other hand, K concentrations obtained in this study (2–55 g/kg) were slightly higher than those reported in the literature (3–32 g/kg).

It was observed from Figure 2 and Figure 3 that the uptake of macronutrients from liquid fertiliser to the plant’s structural parts was fully functional. However, there was no discernible trend in the macronutrients and cations uptake between the plants grown using FoodLift and CLF, except in the case of tomato plant. In the case of the cherry tomato, plants grown using FoodLift appear to uptake higher N and cations (K, Ca, and Mg).

### 3.3. Statistical Analysis

To analyse the macronutrients and cation uptake by lettuce, cucumber, and cherry tomato plants when FoodLift and CLF were used, a detailed statistical analysis was carried out. To determine whether there are any significant differences in the concentrations of macronutrients and cations in the plants grown using FoodLift and CLF, a *t*-test statistical analysis was carried out. Table 3 statistically compares the differences between the selected two liquid fertilisers and identifies whether it is significant (*p* < 0.05), highly significant (*p* < 0.01), extremely significant (*p* < 0.001), or insignificant (*p* > 0.05). As shown in Table 3, out of the 70 analyses for ‘*p*’, only the cations appear to show some significant differences in their uptake by plants between FoodLift and CLF. As observed earlier, there appears to be no differences in the uptake of N and P by the plants which are grown using FoodLift and CLF. These results indicate that FoodLift can replace CLF as a hydroponic nutrient source. Furthermore, it should be noted that experiments with FoodLift went smoothly and there was no odour issue. However, some of the cations and other micronutrients may need to be supplemented while using FoodLift. Further research needs to be conducted to determine the type of nutrients which need to be supplemented and to estimate their quantities. Additional research is also required to monitor the significance of organic content in FoodLift on plant growth and the health and safety of hydroponic operators.

## 4. Conclusions

This study evaluates the nutrient contents of lettuce, cucumber, and tomato produced using food waste-derived liquid fertiliser (FoodLift) and commercial liquid fertiliser (CLF) in a hydroponic system. The organic liquid fertiliser was extracted from food waste (FW) using the procedure reported by Siddiqui et al. (2021). The authors have earlier reported the use of FoodLift as a hydroponic nutrient solution and compared its performance with CLF. In this study, nutrient concentrations, such as nitrogen (N), phosphorus (P), calcium (Ca), potassium (K), and magnesium (Mg) were determined for the fruits and/or leaves, roots, stems, and stalks of lettuce, cucumber, and cherry tomato plants. These concentrations were determined for both the plants which were grown using FoodLift and CLF. Comparison of concentrations in the plants grown using FoodLift and CLF indicated that the plants have similar concentrations of N and P in their various structural parts, except in the case of cherry tomato plants. For cherry tomatoes, the N concentration in the roots and stalks of the plant grown using FoodLift was higher than the concentrations found in the plants grown with CLF (*p* < 0.05). In terms of concentrations of N and P, lettuce had higher concentrations (50–260 g/kg and 11–88 g/kg, respectively) than cucumber and cherry tomato plants (1–36 g/kg and 4–33 g/kg, respectively). However, in the case of cations (K, Ca and Mg) there appear to be some differences, which are statistically extremely significant. These differences appear to vary depending on the cation and the structural component of the pant. This aspect needs to be further investigated. The above results indicate that the FoodLift has the potential to replace CLF for producing lettuce and cucumber in a hydroponic system. However, in the case of cherry tomatoes, FoodLift produced a significantly lower yield than CLF. Nevertheless, FoodLift has the potential to be used as a source of fertiliser for hydroponic systems. Use of FoodLift will help to improve the sustainable use of food waste by recycling its essential nutrients; currently wasted into the environment resulting in eutrophication and other related pollution. It will also provide an alternative source of nutrients for an intensive food production system using hydroponics to meet the growing demand for fresh vegetables. Thus, this paper demonstrated the potential for applying circular economy principles to manage food waste, leading to a “circular economy of nutrients”.

## Figures and Tables

**Figure 1 foods-12-00719-f001:**
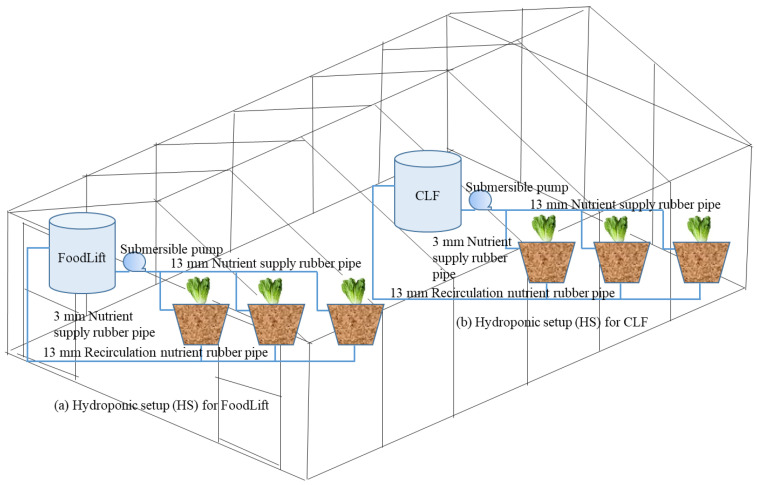
Experimental setup for hydroponic experiments.

**Figure 2 foods-12-00719-f002:**
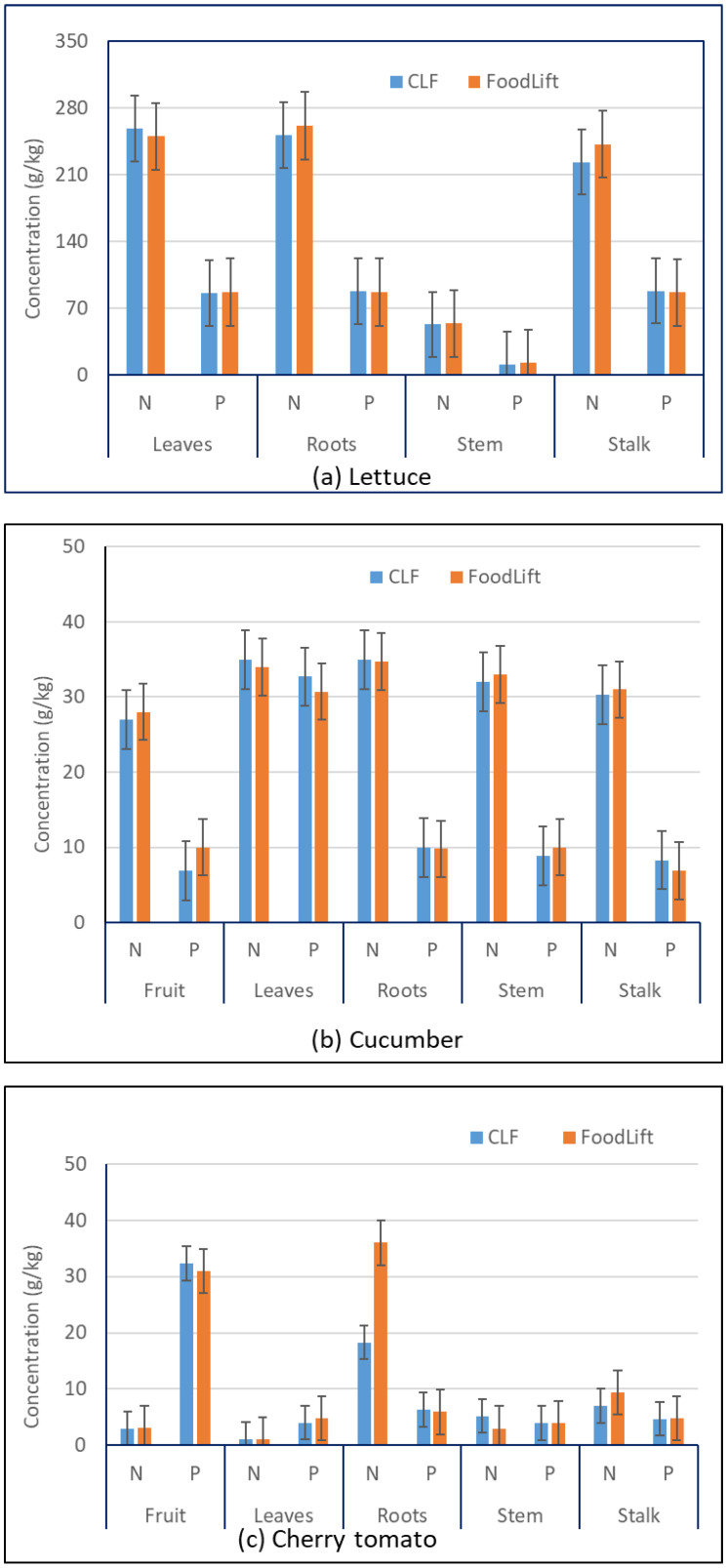
Nitrogen (N) and phosphorus (P) in parts of (**a**) lettuce, (**b**) cucumber, and (**c**) cherry tomato plants (error bars indicate standard deviation (SD) values).

**Figure 3 foods-12-00719-f003:**
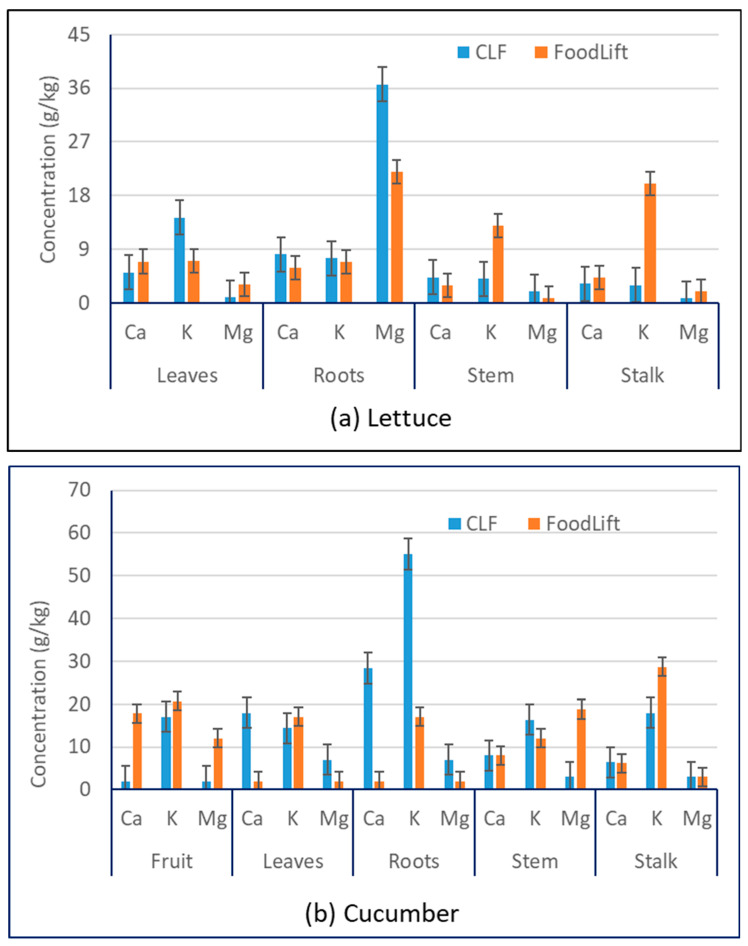
Calcium, potassium, and magnesium concentrations in different structural parts of (**a**) lettuce, (**b**) cucumber, and (**c**) tomato plants (error bars indicate standard deviation (SD) values).

**Table 1 foods-12-00719-t001:** Nutrient concentrations in different structural parts of the plant from literature.

Plant	Structural Part	N, g/kg	P, g/kg	Ca, g/kg	Mg, g/kg	K, g/kg	Reference
Lettuce	Leaf	3	8.6				Delaide et al. [36]
	Leaf		5.7–6.7				Sublett et al. [37]
Continental cucumber	Leaf	25–45	45–40	25–50	3–15	3–7	Parker et al. [38]
	Root	37	10	11.5	2.9	27.6	Sung et al. [39]
	Fruit	27.3	9.6	2.1	2	21.8	Colla et al. [40]
	Stem	32.3	9.7	26.9	4	61	Sung et al. [39]
Cherry tomato	Leaf	33.4	3.7	21.3	5.4	20.7	Maia et al. [30]
	Root	24.5	6.3	4.3	6.9	18.7	Maia et al. [30]
	Fruit	29	22			31.9	Maia et al. [30]
	Stem	28.8	4.4	8.4	3.8	22.2	Maia et al. [30]

**Table 2 foods-12-00719-t002:** Initial water quality parameters and nutrient concentrations in hydroponic feed solutions.

Parameters	Unit	Lettuce		Cucumber/Cherry Tomato			
		FoodLift Feed Solution	CLF Feed Solution	Growing Stage		Flowering Stage	
				FoodLift Feed Solution	CLF Feed Solution	FoodLift Feed Solution	CLF Feed Solution
pH		5.5 ± 0.3	5.5 ± 0.3	5.5 ± 0.3 ^a^ 6.5 ± 0.3 ^b^	5.5 ± 0.3 ^a^ 6.5 ± 0.3 ^b^	5.5 ± 0.3 ^a^ 6.5 ± 0.3 ^b^	5.5 ± 0.3 ^a^ 6.5 ± 0.3 ^b^
Water temperature	°C	24	24	24	24	24	24
Dissolved oxygen (DO)	mg/L	>7	>7	>7	>7	>7	>7
N	mg/L	96	96	96	96	160	160
P	mg/L	38	40	38	40	64	160
K	mg/L	260	460	260	460	433	288
Ca	mg/L	81	70	81	70	162	140
Mg	mg/L	9	32	9	32	18	64

^a^ pH for lettuce and cucumber plants; ^b^ pH for cherry tomato plants.

**Table 3 foods-12-00719-t003:** Fresh matter (FM) yield (g/plant) obtained for cherry tomatoes.

	Plant 1	Plant 2	Plant 3	Plant 4	Average (SD)
FM yield for fruits grown using FoodLift, g/plant	0	0	23	102	31 (48)
FM yield for fruits grown using CLF, g/plant	33	339	308	428	277 (170)

SD: standard deviation.

**Table 4 foods-12-00719-t004:** Statistical summary of *p* (one-tail) values to determine whether the macronutrients and cation concentrations differ significantly between plants/produce grown using CLF and FoodLift.

Produce	Parts	Macronutrients and Cations				
		N	P	Ca	K	Mg
Lettuce	Leaves	0.39	0.35	0.048 *	0.0007 ***	0.016 *
	Roots	0.35	0.14	0.012 *	0.16	0.0008 ***
	Stem	0.38	0.178	0.001 **	0.0005 ***	0.003 **
	Stalk	0.3	0.41	0.044 *	0.00006 ***	0.0013 **
Cucumber	Fruits	0.47	0.3	0.001 **	0.060	0.0017 **
	Leaves	0.43	0.28	0.0007 ***	0.107	0.000002 ***
	Roots	0.5	0.42	0.003 **	0.0006 ***	0.0104 *
	Stem	0.48	0.48	0.5	0.0167 *	0.0055 **
	Stalk	0.44	0.15	0.35	0.043 *	0.226
Tomato	Fruits	0.33	0.15	0.28	0.00005 ***	0.003 **
	Leaves	0.46	0.042 *	0.33	0.053	0.0068 **
	Roots	0.0007 ***	0.22	0.0002 ***	0.0019 **	0.0047 **
	Stem	0.012 *	0.26	0.005 **	0.0004 ***	0.012 *
	Stalk	0.038 *	0.44	0.028 *	0.000003 ***	0.022 *

* indicates significant differences in the concentrations of nutrients and cations present in the plants/fruits grown using FoodLift and CLF (*p* < 0.05). ** indicates highly significant differences in the concentration of nutrients and cations present in the plants/fruits grown using FoodLift and CLF (*p* < 0.01). *** indicates extremely significant differences in the concentration of nutrients and cations present in the plants/fruits grown using FoodLift and CLF (*p* < 0.001). All statistically significant differences are highlighted with the blue background table cells.

## Data Availability

Data is contained within the article or Appendix A.

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
