# Peer review of "A Food Waste-Derived Organic Liquid Fertiliser for Sustainable Hydroponic Cultivation of Lettuce, Cucumber and Cherry Tomato"

_foods, 2023, doi:10.3390/foods12040719_

Round 1
Reviewer 1 Report
The paper "A Food Waste Derived Organic Liquid Fertiliser for Sustainable Hydroponic Cultivation of Lettuce, Cucumber and Tomato" present some relevant results for readers of Foods journal. However, some improvements are required. Namely:
- The description of the procedure of the Organic Liquid Fertiliser (FoodLift )should be included in the paper. Support this only in the reference [33] is not appropriate for the readers. Of course, it should be given only a brief description.
- The discussion of “statistical and Principal Component Analysis” must be improved. In particular Fig.8.
- pg 11, the citation of Fig.7 should be Fig. 8
- It must be mentioned in the paper, if the safety of the FoodLift is guaranteed since it was obtained from food waste. What are the problems related to storage? There are bad odours involved? Etc.
- In the conclusions, it is stated that “However, in the case of cations (K, Ca and Mg) there appear to be some differences, which are statistically significant.”
But if there are several statistically relevant differences, is it possible to conclude that “The above results indicate that the FoodLift can easily replace CLF for producing lettuce, cucumber and tomato in a hydroponic system.” ?
- There are several problems in the references that need to be fixed (for example: 2 "Renju Babu, P. M. P. V. E. R. R., 2021. Strategies...." what is the reason for so many letters (P. M. P. V. E. R. R.)? There are many similar situations.]
Author Response
Our sincere thanks to Reviewer 1 for valuable comments. Attached is the response to Reviewer 1 comments.

Reviewer 2 Report
The authors have performed a work on applying food waste liquid fertilizer as a sustainable means for hydroponic cultivation of several crops. the work has shown the potential of applying food waste for growing crops in an efficient manner to help elevate the food waste management problems and to allow better circular economy.
The abstract can include some important data and points that are significant to the discovery within this work.
The introduction needs to include more on the problem statement and suggestions on how food waste can be used for other applications and why it is beneficial for hydroponic cultivation. Some good reads include:
-Transformation of biomass waste into sustainable organic fertilizers
-The effects of biofertilizers on growth, soil fertility, and nutrients uptake of oil palm (Elaeis guineensis) under greenhouse conditions
-Role of plant growth promoting rhizobacteria in sustainable production of vegetables: current perspective
the distinction of cherry tomato and tomato can be defined more clearly by stating the relevant tomato type throughout the manuscript. the brand of commercial liquid fertilizer can be mentioned too and also the type of food waste expected in the FoodLift liquid fertilizer
Does the work also investigate on the growth parameters of the crops, such as height, weight, colour, composition which are important to know if the liquid fertilizer from waste can be a useful substitute.
For nutrient content, authors can explain about the possible types of food combinations used in the food waste liquid that could have contributed to the respective nutrient content
The significance of the cations in plants growth can be explained further and which of them would show better effect on the growth of these crops.
The figures blend the number values and may interlap, i suggest the author to redraw the figures 2-6 to present them in a more organized manner.
After detecting the possible composition and cations in the three crops, how can the authors relate these nutrient content to the overall growth and fertilization aid provided by the fertilizers?
What does figure 8 intend to show? if it is not too significant, it can be removed
Avoid references in the conclusion section.
Author Response
Our sincere thanks to Reviewer 2 for valuable comments. Attached is the response to Reviewer 2 comments.

Reviewer 3 Report
Comments and Suggestions for Authors
Compared with commercial fertilizer, this research studied the used fertilizer derived from food waste as nutrients for lettuce, cucumber and tomato growing.
Below are my comments and suggestions.
- The abstract should state the numerical results.
- I recommend introducing some recent references; only two references from 2022 have been cited.
- In the paragraph at lines 90 -92 was specified that “. Siddiqui et al. [34] reported the yields of lettuce and cucumber using FoodLift as liquid fertilizer and found that the yield was similar to the one obtained from commercial liquid fertilizer (CLF).” please specify the obtained yield.
- It is essential to underline the novelty of this paper compared with the already published article, reference 34; the paper is the same (ex. Figure 1 is the same as Fig 2 from the reference [34]. Please clarify.
- Please give the references, standards, etc., for Analytical methods presented in subchapter 2.3.
- I recommend to improve the clarity of the Figures.
- It was good to have some data about on the transfer of cations from fertilizers into the plant, to calculate transfer indices, etc.
Author Response
Our sincere thanks to Reviewer 3 for valuable comments. Attached is the response to Reviewer 3 comments.

Round 2
Reviewer 2 Report
The authors have addressed the comments appropriately
Reviewer 3 Report
The paper was substantially improved, but I have a remark.
1 Please renumbered the reference [44] which appear at line 94 after reference [32].